# Mushroom Nutrition as Preventative Healthcare in Sub-Saharan Africa

**Tito Fernandes [1,\*]** , **Carmen Garrine [1,2]**, **Jorge Ferrão [3]**, **Victoria Bell [4] and Theodoros Varzakas [5]**

1.  CIISA, Faculty of Veterinary Medicine, University of Lisbon, 1300-477 Lisbon, Portugal; carmen.garrine@fmv.ulisboa.pt
2.  Faculdade de Veterinária, Universidade Eduardo Mondlane, 3453 Maputo, Mozambique
3.  Vice-Chancellor's Office, Universidade Pedagógica, Rua Comandante Augusto Cardoso, 1260 Maputo, Mozambique; reitoria@up.ac.mz
4.  Faculdade de Farmácia, Universidade de Coimbra, 3000-548 Coimbra, Portugal; victoriabell@ff.uc.pt
5.  Department Food Science and Technology, University of Peloponnese, 24100 Antikalamos, Greece; t.varzakas@uop.gr
\*   Correspondence: profcattitofernandes@gmail.com; Tel.: +351-91-992-7930

**Abstract:** The defining characteristics of the traditional Sub-Saharan Africa (SSA) cuisine have been the richness in indigenous foods and ingredients, herbs and spices, fermented foods and beverages, and healthy and whole ingredients used. It is crucial to safeguard the recognized benefits of mainstream traditional foods and ingredients, which gradually eroded in the last decades. Notwithstanding poverty, chronic hunger, malnutrition, and undernourishment in the region, traditional eating habits have been related to positive health outcomes and sustainability. The research prevailed dealing with food availability and access rather than the health, nutrition, and diet quality dimensions of food security based on what people consume per country and on the missing data related to nutrient composition of indigenous foods. As countries become more economically developed, they shift to "modern" occidental foods rich in saturated fats, salt, sugar, fizzy beverages, and sweeteners. As a result, there are increased incidences of previously unreported ailments due to an unbalanced diet. Protein-rich foods in dietary guidelines enhance only those of animal or plant sources, while rich protein sources such as mushrooms have been absent in these charts, even in developed countries. This article considers the valorization of traditional African foodstuffs and ingredients, enhancing the importance of establishing food-based dietary guidelines per country. The crux of this review highlights the potential of mushrooms, namely some underutilized in the SSA, which is the continent's little exploited gold mine as one of the greatest untapped resources for feeding and providing income for Africa's growing population, which could play a role in shielding Sub-Saharan Africans against the side effects of an unhealthy stylish diet.

**Keywords:** food insecurity; mushroom nutrition; poverty; health promotion; health foods



## 1. Introduction

The basic human Right to Food, a 73-year-old commitment of all countries, should have guaranteed each living person to be exempt from hunger, which is mainly generated and perpetuated by human decisions and the dependency on international trade agreements. Universally, the concept has evolved to Right of Adequate Food interlinking policies on agriculture and nutrient requirements with fields such as environment, climate, energy, education, social–economy, and marketing.

Despite regular and record numbers of national and international campaigns, programmes, initiatives, global development goals, universal declarations, technical and scientific articles, and books, Sub-Saharan Africa stands as the world's most food-insecure region. Albeit advances, hunger, food insecurity, and under nutrition still prevail as a serious hazard and the United Nations Zero Hunger Challenge by 2030 is in doubt and probably unachievable.

Agriculture and Fisheries, and associated sectors, are the main sectors of occupation for the majority of African people. Agriculture, sea, and river resources are the impelling cause of economic reform in Africa, since it bears the world's largest unfarmed arable land and marine assets, employing a large fraction of the population [1]. Nevertheless, SSA is a net food importer that is dependent on most agricultural and agro-food sectors, namely maize, rice, and wheat. This situation was aggravated in the last four decades, mainly due to rapid population growth, persistent economic inequality, climate change threats, and even claims of the legacy of colonialism.

In order to obtain a SSA food sovereignty, further to the general need to increase the production and productivity of cereals, untapped traditional and native food crops with expected nutritional attributes remain to be extensively researched and considered, having massive potential to improve the agri-food and fisheries' value chains [2].

Smallholder farmers and fisherman, with a valuable central role to play, have been struggling on a subsistence level often with no community-control and biodiversity-based food systems. There is no panacea for these issues, and the first movement must be on investments in agricultural and fisheries infrastructures and extension services, as smallholders are key actors in food security and in poverty reduction [3].

The leverage of agriculture for food and nutrition security is a means to improve human health and dietary patterns toward increasing agricultural diversity and ensure a balanced diet. However, there is little health research on diet quality based on what African people consume, and a rigorous evaluation of for example universal micronutrient supplementation effects from international aid programmes is extremely rare.

The practice for the past decades of vast and well-intended international aid, even from the United Nations WFP, to the region to curb food insecurity has been unsustainable, ineffective, with unintended consequences and may even ultimately cause harm. It is essential to link aid effectiveness to catalyse development strategies with a longer-term focus.

In SSA, comprising some 44 countries, despite poverty, chronic hunger, food insecurity, movement to renewal, and the arrival of new foods and eating habits, the traditional food choices have luckily prevailed and been considered beneficial in relation to health outcomes and sustainability [4].

However, it is changing, since with no trade agreements, many international organizations and food companies dump their products to gain market share in SSA. Since the 1960s, the African people have consumed increasing amounts of processed food.

The global food system is very complex and influenced by many different inputs, including farming, economics, politics, environment, transport, storage, and consumers; it must entail long-term dimensions on sustainability. These factors are aggravated in SSA, the second world region with the highest prevalence of under-nutrition as well as inadequate incomes or other resources.

Malnutrition is still one of SSA's primary concerns for enhanced human development. Due to inadequate dietary intake and lack of nutritional knowledge, there is a frequent concurrence of both under-nutrition and over-nutrition in the same population across the life course probably due to unbalanced diets or diseases [5]. There are multiple reasons for malnutrition and promoting actions must be multi-sectorial, although quite complex to coordinate.

In general, despite indications that Africans are smoking less and having more physical exercise than in developed countries, when food is available, the African diet rivals the healthy Mediterranean diet. African cuisine is a healthy way of cooking and may become an example if food diversity is enhanced. However, in African urban areas, with the growing acceptance of "Western" eating habits, one can expect more non-communicable diseases or chronic diseases (e.g., diabetes, cardiopulmonary diseases, cancer) for which African healthcare systems are unprepared [6].

Among many possible initiatives to improve food and nutrition security in SSA, it is important to identify consumer habits in each region. Ideally, a guideline needs to be specifically designed for each of the main six African regions or even per country.

The incorporation of African indigenous foods into the existing diet must be incentivized. African "superfoods" and other functional foods and beverages, with traditionally proven major health benefits, should be encouraged.

Knowledge about what is eaten is essential to dietetics and food science as well as for biodiversity, agriculture production, and the food industry. Food pattern recommendations usually limit the intake of salt, sugar, and saturated fats and are normally derived from established dietary guidelines. While these have been well structured in developed countries [7], only very few SSA countries have achieved this stage.

An overview of the African foods and ingredients and the importance of establishing national dietary guidelines that apply to each country or region are discussed. Contrary to what is known from ancient Asian civilizations, the ethnomycological knowledge of useful African mushrooms is scant. Furthermore, gut microbiota from SSA people have different and specific profiles, which need to be studied in order to match and determine their nutrient requirements. Since most rural SSA small farmers operate traditional subsistence lifestyles, it is important to evaluate their microbiota profile and role as well as the widespread antibiotic use [8].

This article is a contribution to the debate about the effectiveness of dietary interventions for African rural development, integrating agricultural interventions for food security with those for poverty reduction while shedding light on mushrooms as novel food source with health-promoting foods and a possible contributor to the African diet, income, health, and livelihood. Our review has several clear limitations, most importantly the lack of accurate data on food production and of epidemiological studies in SSA.

## 2. The African Diets

The diverse nature of African cooking has fantastic elements of different cultures: Arab, Black African, European, and Asian. African eating and drinking habits are significantly different in each African region. Presently, there are some five to six main African regions (Figure 1) and there are not many studies on food consumption patterns of the African people per country or among the 54 sovereign countries [9].

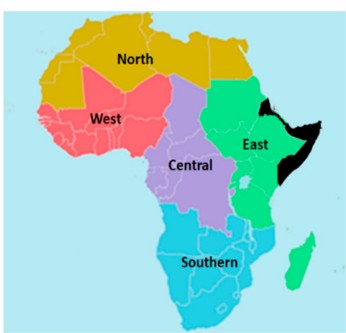

**Figure 1.** Main African regions.

Africans, namely rural communities, perform more physical exercise and do not eat many ultra-processed foods such as salty fatty packaged snacks, soft fizzy drinks, sweetened breakfast cereals, and instant convenience foods, which are ultra-processed and nutritionally unbalanced [10].

The traditional African diet comprises more wholesome and healthful foods rather than pre-treated food (Figure 2). In general, the defining characteristics of the traditional African cuisine are rich herbs and spices, fermented foods and beverages, and healthy and whole ingredients [11]. In comparison with other continents, very little meat, fish, and poultry is generally consumed in Africa [12].

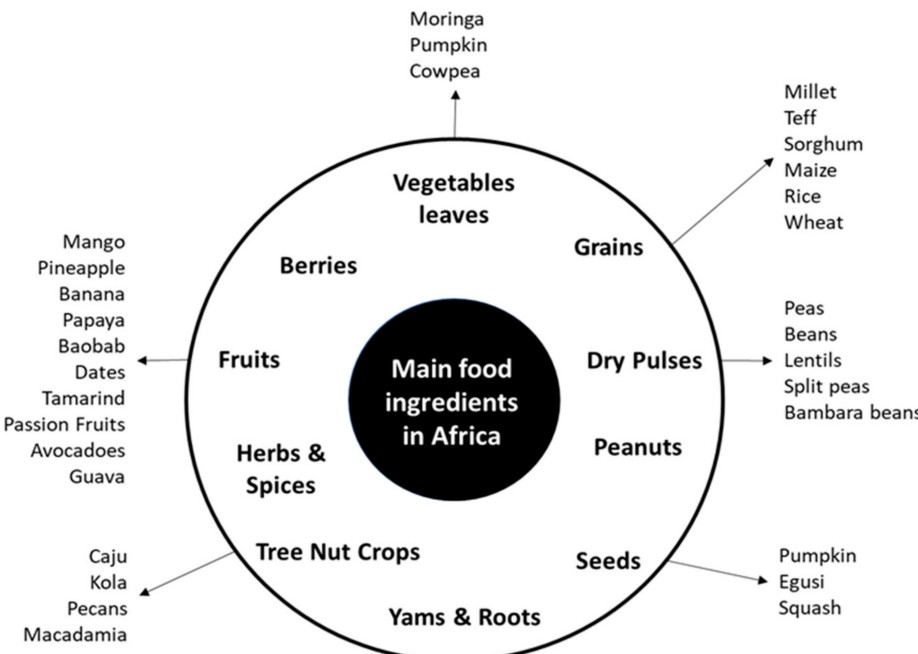

**Figure 2.** Main food ingredients in Africa.

It is considered that people in West Africa (Mali, Chad, Senegal, and Sierra Leone) enjoy healthier diets than their counterparts in the United States, the United Kingdom, Japan, or Canada [13].

Some select African high-nutrition foods, "superfoods" worthy of this title, have been identified, with high concentrations of essential nutrients such as phytonutrients, vitamins, minerals, enzymes, and antioxidants, although no single food holds the key to good health or disease prevention. They include moringa, baobab, yams, teff grain, leaves of kenkiliba, sesame, fonio grain, artemisia, tamarind, hibiscus, coconut, pumpkin, and amaranth leaves (Figure 3).

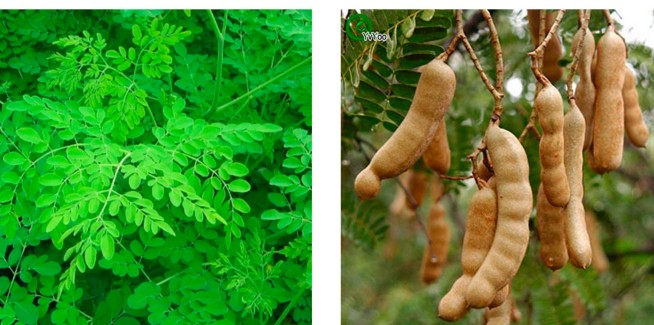

**Figure 3.** Moringa leaves and tamarind fruit, widely used as food and for medicinal purpose.

Other foodstuffs with good nutrient counts and health properties comprise local darky leafy greens, green tea, legumes/pulses, nuts and seeds, yoghurt, garlic, ginger, curcumin, avocado, sweet potatoes, seaweeds, and mushrooms. Cooking oils from fruits (e.g., palm), seeds (e.g., sunflower), and nuts (e.g., peanut) are widely used, while olive oil is also important but in northern Africa where Tunisia and Morocco are considered the largest producers in the world.

Meat, fish, seafood, and traditionally produced fermented dairy products (e.g., yogurt, cromwo, boeber, alouda, amasi, and leite azedo) are often used as a garnish, prepared with cooking oil, tomatoes, onions, salt and spices, poured into a mash or porridge made from cereal or cassava flour. Beef, goat, chicken, eggs, and mutton are quite expensive in SSA,

so these foods are reserved for special days. However, fish and seafood are abundant in coastal regions, rivers, and lakes.

Most countries should periodically review their multi-sectorial nutrition strategies and consider both more effective nutrition-specific intervention approaches (e.g., supplementation, fortification) and/or a shift to actions at the level of underlying causes (e.g., promoting optimal breastfeeding and complementary feeding, diet diversification). We have previously considered the issue of monotonous diet in SSA, lack of diversification, and fortification as a matter of business as much as science [14].

## 3. African Dietary Guidelines

Modern nutritional science is surprisingly young, and "Food-Based Dietary Guidelines" are suggestions based on science knowledge in the form of instructions for healthy eating. Designed for information to consumers, customers, and technical advisors, they must be suitable and relevant for each country, culturally appropriate, and easy to implement. Moreover, they should be harmonious, comprehensible, and memorable [15].

Authorities use a spectrum of schemes from optional to compulsory. The present global food regulatory framework is confusing and limiting, and specific measures exist for certain claims [16]. Many challenges still remain regarding the establishment of dietary guidelines integrating education, agriculture, health, environment, and industry.

Many developed countries have established dietary guidelines, but implementation plans are often not comprehensive enough for consumers. Dietary guidelines around the world are presented in different configurations, exhibited as pyramids, plates, baskets, texts, circle graphs, diagrams, and tables, yet they are similar in terms of content, giving consumers a number of advised food groups and daily servings to maintain optimum health [17].

In Africa, food-based dietary guidelines have been established in Benin, Kenya, Namibia, Nigeria, Seychelles, Sierra Leone, and South Africa, the latter in 2003. Food is a particularly sensitive commodity, but the majority of SSA countries did not develop their own food guidelines [18], and in general, the approach of the legal systems has been broadly consistent with the benchmark in international trade agreement channelled through the *Codex Alimentarius* Commission since 1963 [19].

In African history, most rural life has been devoted for household production and the procurement and preparation of foodstuffs, often low in nutrients, while food scarcity has constituted a major threat to survival. Unless the food quality (i.e., safety and nutrient composition) that Africans eat is addressed, the continent will not be able to address under-nutrition, obesity-related diseases, and even mental health.

We have enhanced the fact that some major nutrient contents of foods are reasonably well characterized, and their required levels of intake calculated. However, the subject becomes quite complex when accounting for the active bioavailability of the dietary compound rather than the dose ingested [20]. Furthermore, there are naturally occurring ca. 100,000 phytonutrients in plants, which are considered non-essential for growth and development but essential for lifetime good health [21].

We have previously reviewed several global dietary guidelines and noticed that with a few exceptions, fermented foods and mushrooms are generally absent as a recommended category of food for daily intake in Food Guides, reflecting a failure to appreciate the benefits resulting from these foods [17].

Current food systems in African agriculture, fisheries, and animal production, an outcome of a historic development pathway, are unsustainable with no diversification and no adequate integration of indigenous products [22,23]. Nevertheless, Africans eat starchy foods in the form of minimally processed or whole grains, legumes, beans, roots, and yams, rather than refined starches and sugary products with the benefit of a high carbohydrate intake supplying 55–75% of total dietary energy [24].

The majority of African cuisines have a different starch base (sorghum, millet, maize, teff, rice, sweet potatoes, cassava, and yams) because they supply plenty of calories. Starches are more filling, as they mislead the body and brain into feeling satiated [25]. Usually, it is

complemented with margarine or oil, supplying an extra source of energy but few essential nutrients where fried onions, garlic, and tomatoes make a basic curried sauce [26].

Inadequate nutrition, whether associated with deficiency disorders or chronic diseases, is embedded in impoverishment and neediness [27]. In the poorer regions of SSA, micronutrient malnutrition exists wherever there is undernourishment due to food shortages, and it is likely to become common where diets lack diversity, even in conjunction with sufficient energy intake [28].

*Role of Mushrooms in the Dietary Guidelines*

Foods supplying proteins in dietary guidelines have been organized under the concept of being either animal or plant based while other rich protein sources, i.e., mushroom-derived, has been relatively neglected [29]. The intake of mushrooms per day has been very low worldwide and regarded as a lavish delicacy; however, their inclusion adds essential shortfall micronutrients and bioactive components.

Mushrooms have a unique and key nutrient profile supporting the recommendation of lower energy intake and sodium, and they are uniquely high in vitamin D and protein as well as low in fat. Mushrooms can be supplied fresh, as biomass dietary supplements, or as extracts. However, the extracts are considered medicinal nutraceuticals and not as foods or dietary supplements, and the legislation regulating these dietary supplements remains unclear due to the fact that they can be considered as foodstuffs and/or medicinal products depending on various factors.

Below, we propose a graphic design of a general Food Guideline for Sub-Saharan Africa with the inclusion of indigenous products (Figure 4). It is an attempt since a sound or comprehensive Sub-Saharan African Food Guide Pyramid should be broader and include other information related to food safety, number of meals, amount of (un)refined foodstuffs, processing stages, access to potable water, food traditions, fermented food and beverages, salt and sugar consumption levels, methods of cooking, sociocultural habits, and even creed and religious faiths.

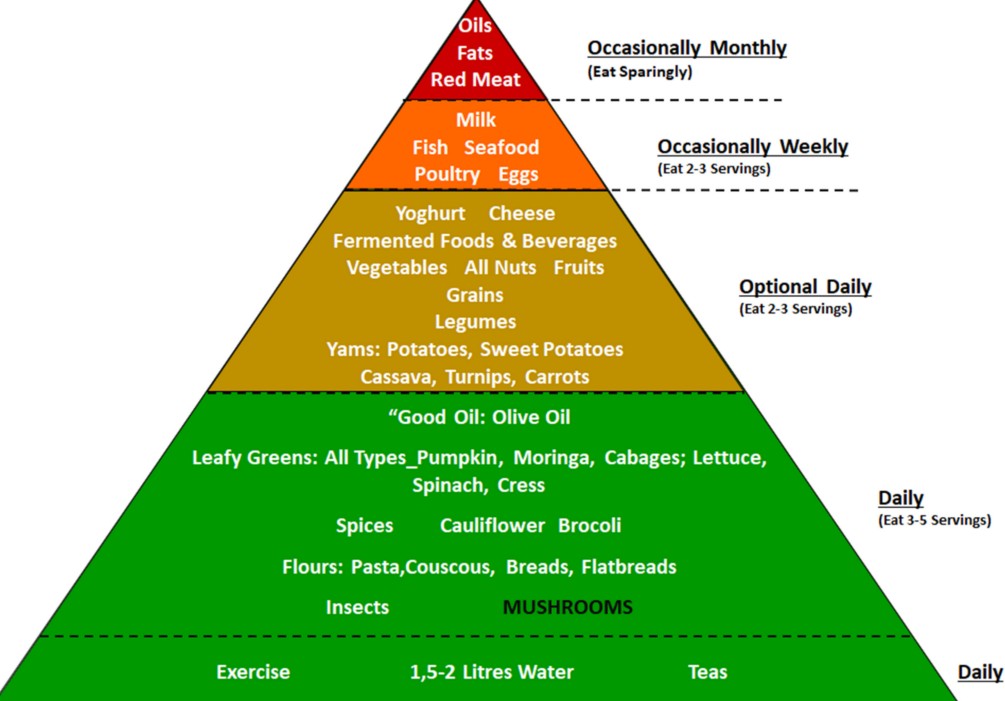

**Figure 4.** The Sub-Saharan Africa Food Guide Pyramid. There are variations in amounts recommended from different food groups. A "serving size" is a standard amount of a food, such as a cup or a spoon, but they are not fixed recommendations. It is recommended to drink a lot of water and teas, exercise physically daily, and spare on salt, sweets, candies, processed foods, squash drinks, and fizzy beverages.

We note the specific inclusion of mushrooms as a valuable food source of essential nutrients and bioactive components, namely protein and medicinal bioactive elements that have been in the past overlooked in global dietary guidelines.

The term "portion" means the amount of a food selected for a refection or snack. A portion size can vary from meal to meal, depending on energy concentration, whereas "serving size" is a measured amount of food usually recommended by the food manufacturer or an external agent. There is room for all foods in a well-balanced, healthy diet, but some should be eaten less often and in smaller portions than others. Appropriate amounts eaten takes practice and depends on the individual.

Even considering developed countries, only very recently there has been a pioneer recommendation of adding a serving (84 g/day) of mushroom mixtures to USDA Food Patterns [30].

In SSA, it is imperative to promote foods with a high ratio of micronutrients to energy content, diversifying food sources, moving away from the concept of food groups and adopting the use of local names for common foods and beverages. Each country, and probably each individual region, must design specific healthy eating patterns [31,32].

## 4. African Mushrooms

Mushrooms that have existed for centuries have spread all over the world, and their distribution in SSA has not been completely surveyed yet. Mushrooms are vitally important to human life and of primary importance for the environment as global decomposers of nature organic matter and recyclers in most ecosystems, and they have been used in SSA since the Palaeolithic period (5000–9000 BC), where their application has been historically related to spiritualism [33].

Mushrooms (aboveground macrofungi) engage central roles in SSA tropical forest ecosystems feeding on non-living organic matter and interacting with trees, and they can influence carbon and nutrient cycling and the maintenance of biodiversity. Truffles (subterranean macrofungi) from arid-land ecosystems enhance the capacity of their host plants to resist dry spells.

Fungi also help in the growth of most plants by developing mycorrhizal symbiotic associations, enhancing crop productivity and aiding to increase access to water and minerals for plants, and tolerance to stressful conditions. Zimbabwe, Swaziland, Namibia, South Africa, Malawi, Benin, and Ghana are the leading mushroom-producing °countries. The production of mushrooms on a small or large scale is quite simple, and good examples can already be seen in Cameroon, Congo, Ivory Coast, Kenya, and Namibia, where 10 kg of fresh mushrooms per square meter and more is achieved.

If fungi are to be the subject of a thriving application in industry and biotechnology in Africa, much more research and development is needed, and consequently mycological education. Some countries (e.g., Zimbabwe) have increased prominence of fungi in the primary school curriculum and awareness of edible and toxic mushrooms. Malawi has a Fungi Farm where children have the opportunity to learn about how the conservation of nature and ecology is key to how we can all live in harmony.

However, most SSA countries did not release guidelines or enact legislation in order to ensure the safe commerce of wild mushrooms due to food safety concerns, and present legislations do not yet mention macro fungi let alone their conservation.

Nutritional information on cultivated species of fungi is extremely vast; however, data on wild edible fungi remain scarce, these being collected for food and as an income [34].

Here, it is highlighted the potential and the current knowledge available on the nutrient, antioxidants, and bioactive components values of some African edible macrofungi underlining the applications of mushrooms as dietary foodstuffs in some major health concerns, and they are also often used for innovative biotechnological, medicinal, and ecological applications [35]. It does not underline their application in complementary folk medicine in this part of the world, but the need to consult indigenous people with this knowledge must be stressed.

The tropical and subtropical regions of SSA are characterized by higher mushroom diversity compared to North Africa. Mushrooms and truffles are considered valuable foods in many cultures being rich source of different types of essential nutrients, and they have been widely studied and reviewed [36,37]. White truffles are considered the most expensive food in the world; however, this bonus income has not been explored in SSA, where less expensive black desert truffles predominate, growing mainly in the rainy season.

No single sector or actor can establish food and nutrition security, and there is the need for a well-coordinated effort among them on cross-sectorial approaches and multi-partner platforms [38]. Cooperative endeavours under a multi-institutional program are required to seize the representative macro fungi species of SSA with a view to update their nutritional and health value [39].

Globally, there are up to 5 million species of mushrooms, of which around 1000 of these species can be found in SSA, and as there is no single tool for their identification, only some 7% have been accurately classified [40]. The poor discovery, identification, and certification of edible and medicinal species of mushrooms in SSA retarded its potential use in nutrition, as tonic, and as medicine [41,42].

Foraging for wild mushrooms in SSA Africa does play a significant role in sustaining their livelihood, but there are very few ethnomycological reports and research on mushroom genetic resources, the cultivation of undomesticated wild mushrooms, protection, and lineage improvement [43].

With the application of molecular methods, it is now possible to identify mushrooms previously considered non-existent in SSA. *Hericium erinaceus* was first reported in Tunisia, but it is common in SSA tropical forests in Ghana, Cameroon, Congo, Madagascar, and currently, it is also successfully cultivated and South Africa, exporting annually over 440 tons [44].

Mushrooms produce a vast set of extracellular carbohydrate-active enzymes and biological active molecules that degrade very complex compounds such as hemicellulose and lignin. The variety of enzymes is dependent on the habitat and specific substrates, so it differs among mushroom species and home ground [45].

Desert mushroom truffles, used for thousands of years in Africa, include genera such as *Phaeangium*, *Terfezia*, *Delastreopsis*, *Balstonia*, *Delastria*, *Leucangium*, *Mattirolomyces*, *Tirmania*, and *Tuber* [46], and they are of considerable interest for ecological reasons because of the low water input or organic matter required for sprouting.

Various types of mushrooms and truffles (Figure 5) are considered as natural biota in the North Africa deserts [47,48] and in South Africa, Namibia, and Botswana [49].

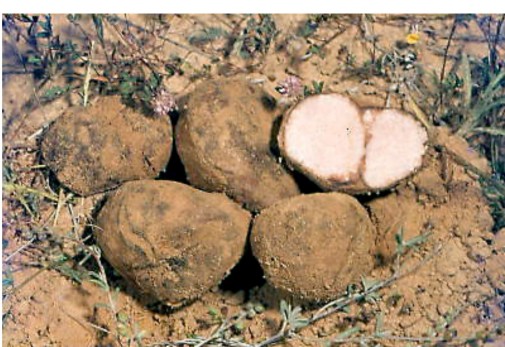

**Figure 5.** African desert truffles genus Terfezia.

Mushrooms belonging to species of *Termitomyces*, *Pleurotus*, *Lentinus*, *Lenzites*, *Trametes*, *Ganoderma*, *Pycnoporus*, *Coriolopsis*, and *Calvatia* have been reported to be used in folk medicine in West Africa. Some popular wild edible and medicinal mushrooms in West Africa include *Schizophyllum commune*, *Lactarius* spp., *Chantarellus platyphyllus*, *Volvariella volvacea*, and *Auricularia auricular-judae* [50,51]. In the Namibe desert, *Agari-*

*cus campestris*, *Calvatia lilacina*, *Coprinus comatus*, *Ganoderma* spp., *Schizophyllum commune*, *Volvariella volvacea*, and *Termitomyces*.

Specific to Africa, there are more than 1000 species from the family Termitidae, constituting 95% of soil insect biomass [52]. Although they are wild, the African mushrooms, and particularly those species associated with termites (*Termytomyces*), are considered "superior" to all other mushrooms [53]. There are about 30–40 mushroom species of *Termytomices* whose cap can reach 1 m in diameter, and most are highly valued as food (Figure 6) [54]. On the other hand, mushrooms of the species *Termitomyces*, *Agaricus*, *Boletus*, *Pleurotus*, *Cantharellus*, *Macrolepiota*, *Ganoderma*, and *Geastrum* have been reported in East and South Africa [37].

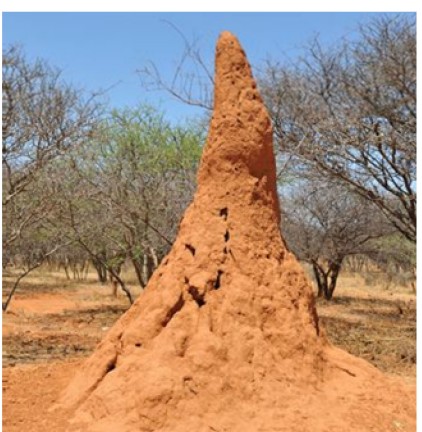 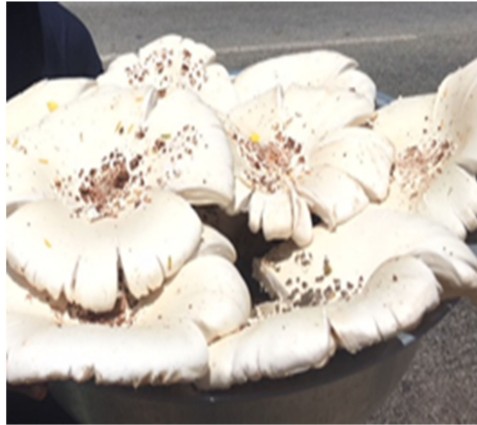

**Figure 6.** A termite and roadside vending of edible mushrooms, Termitomyces mushroom, a traditional source of seasonal income in miombo areas in Mozambique.

The indigenous population on miombo, a savanna type of ecosystem woodlands, relies directly on the timberland for their resources for wood fuel, food and wild fungi. This region extends from East Africa (Burundi, Kenya, Tanzania, Uganda) to the Zambezian region (Angola, DR Congo, Malawi, Mozambique, Zambia, Zimbabwe), where the most predominant mushrooms occurring include *Cantharellus* (15 spp.), *Lactarius* (incl. *Lactifluus*) (14 spp.), *Russula* (10 spp.), and *Amanita* (8 spp.) [55].

Despite its millennial existence and its empirical knowledge, harvesting wild mushrooms is not a well-known concept in Africa due to the threat of being poisonous and sociological impacts (myth, culture, and spirituality) [56,57], while commercial production exists but is still in its early stages.

Mushroom cultivation is a lucrative agricultural process to produce various essential nutrients from plant wastes, requiring a correct combination of compact space, temperate climate, high humidity, and organic substrates residues [58,59].

*Bioactive Compounds of Mushrooms*

Unlike other foods, macro-fungi mushrooms act through major categories of specific bioactive molecules: (1) polysaccharide β-glucans or polysaccharide–protein complexes; (2) triterpenes; (3) polyphenols; (4) alkaloids; (5) metalloids; (6) short-chain fatty acids; (7) enzymes; (8) lectins; (9) nucleotides.

Mushrooms feed on dead plant material, fulfilling an essential role in the carbon cycle while harbouring numerous species with diversity of metabolites of nutraceutical and therapeutic significance [60].

Mushrooms are heterotrophic (do not perform photosynthesis) and reproduce through spores absorbing complex organic compounds from the environment, as they are unable to synthesize their own organic matter [61]. The mycelia, which play important roles in the support and absorption of nutrients, rely solely on carbon obtained from other living organisms, i.e., plants, insects, and even other mushrooms, for growth.

Considerable debate is ongoing on the definitions of nutraceuticals (first coined in 1989 by Stephen L. DeFelice), functional foods (coined in Japan in the early 1980s), innovative food products, with extensive disagreements, and no existing international agreements, which compounds the confusion. We regard nutraceuticals as products, which other than nutrition are also used as alternative for pharmaceuticals.

There is abundant literature, and we have described the value of some edible mushrooms and culinary–medicinal mushrooms, their biomass and extracts, which contain many low molecular bioactive components termed secondary metabolites, since they are formed due to the enzymatic resections of primary substances (amino acids, sugars, vitamins) [62].

The most common secondary metabolites in mushrooms include polyphenols, phenolic acids, quinones, coumarins, groups of flavonoids, stilbenes, hydrolysable and condensed tannins, terpenes and terpenoids, alkaloids, lectins, sterols, lactones, antibiotics, and metal-chelating agents, all of which may activate the cell and humoral immunity, hence increasing resistance to disease [63–66].

The different bioactive polyphenolic compounds act as effective antioxidants based on their excellent ability to scavenge free radicals and act as reducing agents [67]. These large numbers of biological cell components and secondary metabolites have been shown to affect the immune system of the human consumer [68,69].

Lignocellulose is the most abundant natural biopolymer on earth, and its structure being so complex affects its biodegradation and rate-limiting steps in the global carbon cycle [70]. Most species of mushrooms synthesize enzymes that may play important functions in the human organism. The vast list of enzymes in mushrooms include hydrolases, glucoamylase, pectinase, acid protease, endo-1,4-β-glucanases and 1,3-β-glucosidase, esterases, phenol oxidases, polyketide synthase, hemicellulases (glucuronoxylanase, arabinoglucuronoxylanase, and glucomannanase), the ligninolytic system, cell wall lytic enzymes (laminarinase, 1,4-β-D-glucosidase, β-N-acetyl-D-glucosaminidase, α-D-galactosidase, xylanases, β-D-mannosidase, acid phosphatase, laccase, lignin peroxidase, manganese peroxidase, polygalacturonase-pectinase, ribonuclease, and many others [71,72]. Thus, mushroom enzymes can break down polyssacharides and the ligninolytic system, releasing compounds or providing a remarkable nutritive value [73].

The *Hericium* fruiting body contains polyphenol oxidases (PPOs), including tyrosinase and laccase, which are strong antioxidant substances [74]. Forest dweller *Hericium erinaceus* strains were first reported in temperate forests of North Africa and Ghana and known to have anti-peptic ulcer activity [44,75–77].

Several studies showed that *Pleurotus eryngii* and *Ganoderma lucidum* can produce laccases, which is a group of enzymes that can confer activity against HIV by inhibiting the reverse transcriptase [78,79].

Superoxide dismutase (SOD) is also present in some mushrooms, and its important physiological role is in the primary cellular antioxidant defense and its potential therapeutic use [80]. Proteolysis is an essential part of many physiological and metabolic processes in all biota, and basidiomycetes mushrooms are valuable sources of proteases used in defense mechanisms of living organisms and in biotechnological processes [81].

Multiple lectins produced by *Flammulina velutipes*, *Pleurotus ostreatus*, and *Ganoderma carpense* were shown to have potent inhibitory activity in vitro toward cancer cells through fungal ribotoxin-based immunotoxins, which are characterized by the ability to irreversibly block protein synthesis in neoplastic cells [82,83].

In addition to the presence of enzymes, we have previously discussed [84] the significance of mushroom low-molecular-weight secondary metabolites (e.g., terpenes, steroids, anthraquinones, and benzoic acid), which can regulate processes such as cell cycle regulation, apoptosis, autophagy, angiogenesis, metastasis, and signal transduction cascades, which are associated with the development of cancer [66,85,86].

Nucleosides and nucleotides, components of nucleic acids, participate in the genesis and retention of energy for basal metabolism, the synthesis of macromolecules, and cell–cell signalling interplay with cell surface protein receptors; this transmits signals and changes

inside the cell and hence the regulation of physiological processes in the human body via the vast array of purinergic and/or pyrimidine receptors [87].

Oxidative damage causes the production of over one hundred known modified nucleotides formed by multi-enzymatic reactions and by spontaneous chemical reactions. Mushroom constituents include these nucleic derivatives (nucleosides, nucleobases, and nucleotides), flavour determinants, and all are involved in several enzymatic reactions as target or as cofactors [88]. They are also used as fingerprint biomarkers to authenticate the mushroom species, and the pyrimidine ring structure, the backbone of the genetic material of DNA, has revealed therapeutic potentials and valuable medical applications, as shown below (Figure 7).

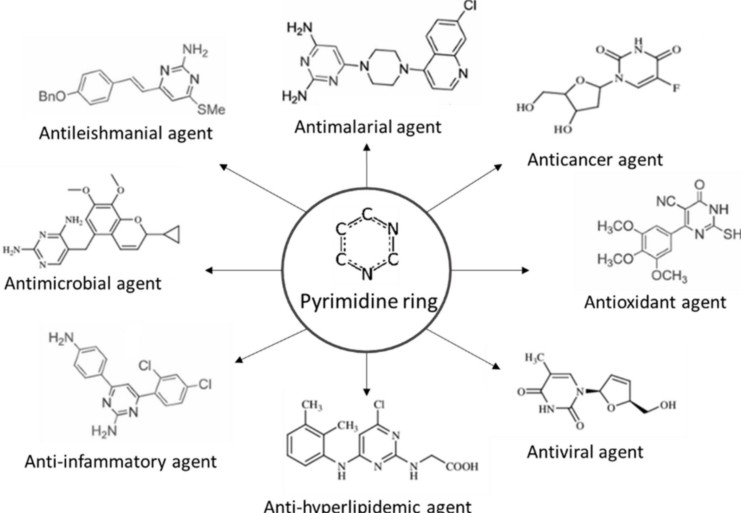

**Figure 7.** Antimicrobial, antioxidant, antimalarial, anticancer, and anti-inflammatory potential functions of pyrimidine derivatives (adapted from [89]).

From shiitake mushroom *Lentinus edodes*, the most studied species of mushroom and the second most consumed in the world, we can obtain lentinacin (eritadenine), a nucleoside compound well known to reduce total blood sugar and cholesterol [90]. DNA are composed of the same four nucleotides, and consumption is considered safe due to the likelihood of transfer and functional integration of DNA from ingested food, even modified, while gut microflora and/or human have developed sophisticated methods to suppress and annihilate exogenous DNA [91]. Nevertheless, there is a practical daily safe limit of nucleic acid intake (ca. 2–4 g) in human adults [92,93].

## 5. Anti-Inflammatory Role of Mushrooms

Inflammation, the cornerstone of pathology, is a complex protective mechanism where the blood flow raises to the area of tissue lesion or infection, which is a necessary part for recovery [94]. Inflammation a healing restorative process; however, it may be adverse also, because it destroys a lot of the fine cells in the process [95].

Based on the need to develop novel therapies, researchers have sought evidence supporting the impact of specific foods on inflammation in the body. Foods that may originate inflammation comprise processed carbohydrates such as white bread and pastries, fried chips, fizzy drinks, red meat, processed sausages, biscuits, desserts, and margarine.

Some foods, mushrooms included, have the capacity to suppress inflammation, but it is unclear how often and how much is needed for this benefit. Following an anti-inflammatory diet, one can fight off inflammation; however, although there is promising research for the impact of some foods, there is no anti-inflammatory miracle food, and although diet is crucial, it is not the single factor [96].

Consuming mushrooms does not necessarily show significant changes on induced inflammatory responses. The result is not surprising, since it would certainly be harmful

to strongly induce or suppress immune function following the ingestion of a commonly consumed food. Mushrooms also have an effect on immune function, but that effect is evident only when the immune system is challenged [97].

Common African mushrooms such as *Pleurotus tuber-regium*, *Termitomyces* spp., *Pleurotus* spp., and *Agaricus* spp. are rich in chitin, which can be hydrolyzed into glucosamine, which is involved in the creation of molecules that protect joints from inflammation [98,99].

Some mushrooms act directly on inflammation. *Cordyceps synensis*, a mushroom that is abundant and diverse in humid temperate and tropical forests at high altitude, not yet reported in SSA, contains a nucleoside compound, cordycepin, that stimulates the production of interleukin 10, an anti-inflammatory cytokine [100].

Wild or cultivated mushrooms, fresh or as dietary supplements, have anti-inflammatory activity occurring through inhibition of the NF-κB signalling pathway, which is a protein complex that controls cytokine production and cell survival, and it is a major transcription factor that regulates genes responsible for both the innate and adaptive immune response [101].

*Poria cocos* mushrooms also contain triterpenes, which have been shown to improve inflammation and treat tumors [102]. Other mushrooms exert an anti-inflammatory effect less directly by quenching damaging free radicals and counteracting oxidation. For instance, Chaga mushrooms (*Inonotus obliquus*) have antioxidant activity, protecting cells against oxidative damage [103,104]. Oyster mushrooms (*Pleurotus ostreatus*) have an antioxidant effect as well [105].

Much of the active polysaccharides, water soluble or insoluble, isolated from mushrooms, can be classified as dietary fibres (i.e., β-glucan, xyloglucan, heteroglycan, chitinous substance) and their glycoprotein complexes [106].

The chemical nature of extracted β-glucan varies from different sources. Cereals and other food contain 2.5–4.5% β-glucans, but these are not capable of controlling immune functions. However, mushroom β-glucans, which consist essentially of a (1,3)-β-linked with small numbers of (1,6)-β-linked side chains, can modulate the autoimmune mechanisms [107].

These biological response modifiers (1,3)-β-glucans interact with the intestinal cell wall and are absorbed into the lymph fluid, where they recruit neutrophils and macrophages and trigger the production of cytokines and stimulate immune function (Figure 8) [108].

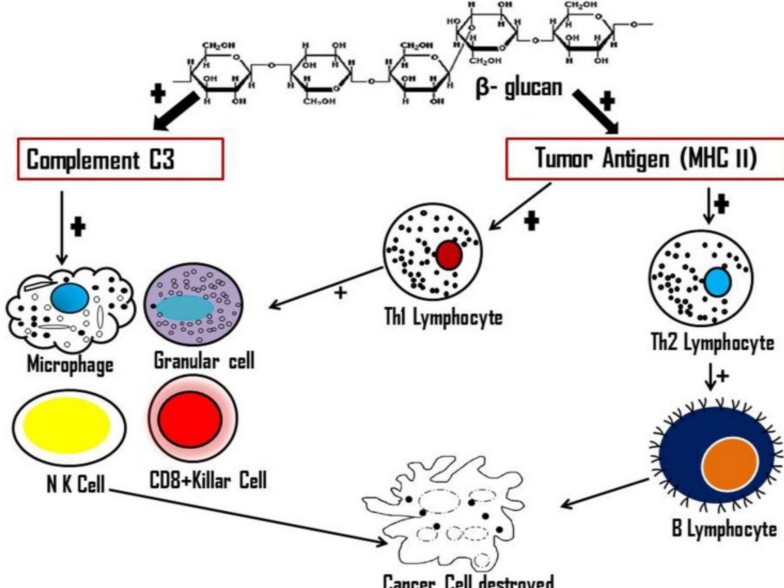

**Figure 8.** Mechanism of antitumor activity of β-glucan bioactive compound [109]. Normally, tumor cells do not express major histocompatibility complex (MHC-II) genes. NK = natural killer. Complement C3 = C3 (C3 deficiency are susceptible to bacterial infection).

Dietary supplements as biomass or extracts derived from the mushroom *Coriolus versicolor* are not foods; they have potential immunomodulating and antineoplastic activities, and they were shown to stimulate the production of lymphocytes and cytokines, such as interferons and interleukins, and they may exhibit antioxidant activities [110,111].

Neuroinflammation is a specialized immune response that occurs in the central nervous system, and it is linked to chronic neurodegenerative disorders (e.g., amyotrophic lateral sclerosis, multiple sclerosis, Huntington's disease, Parkinson's disease, and particularly Alzheimer's), negatively affecting mental and physical functioning being characterized by synaptic dysfunction and a gradual loss of neurons from specific regions [112,113].

Mushrooms incorporate ergothioneine, which humans are unable to synthesize, a unique antioxidant, cytoprotective, and anti-inflammatory derived from food histidine, but which accumulates to high levels in red blood cells and in many other tissues, functioning both as a therapeutic and possibly as a preventative agent of several diseases [114,115].

## 6. The Antiviral Role of Mushrooms

New viruses emerge all the time and can be serious threats to public health. Recently, it was reviewed how mushrooms represent a vast source of bioactive molecules, which could potentially be used as antivirals [116].

A virus is an infectious agent metabolically inert made up of a core of genetic material, either DNA or RNA, and an outer protein and lipid shell, which can only replicate using the host cell mechanisms [116].

Many of the common edible mushrooms and several non-edible mushroom dietary supplements are sources of natural bioactive compounds responsible for the prevention and treatment of viral diseases through their improvement of human immunomodulation (Figure 9) [117]. Numerous previous studies have demonstrated mushrooms as exhibitors of potential antiviral efficacy [118–120].

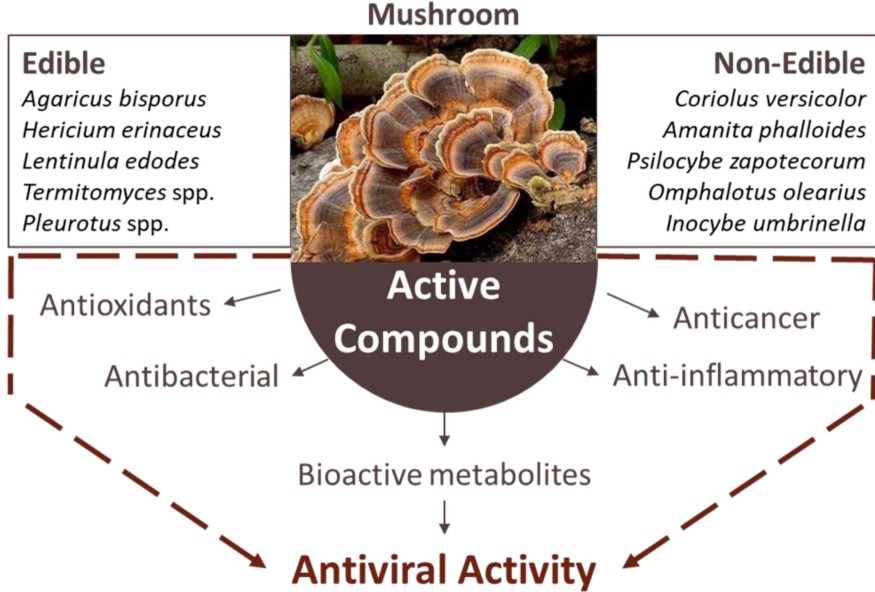

**Figure 9.** With over 400 bioactive compounds, mushrooms have shown a number of antiviral effects, and used as dietary supplements, functional food or medicinal products. They act by blocking virus entry into the cell, by inducing lysis of virus infected cells through activation of NK, CD8+ and T cells, by anti-neuroaminidase increased activity, and by innate immune support.

There are four mushrooms subjected to several clinical studies specifically for fighting viruses [121,122], but the following claims are still considered unsubstantiated at least for COVID-19 prevention and treatment: (1) Ganoderma: shown to kill the Influenza A virus, herpes, hepatitis, and H1N1 strain of the flu; (2) Cordyceps: fighting the Influenza virus by boosting the body NK cell activity as well as other virus-killing cytokines; additionally,

it has been shown to decrease inflammation in chronic asthma and other lung diseases; (3) Maitake: it has shown to actually stop the replication of the virus, which would be very helpful in allowing the body to fight it off without getting too overwhelmed and preventing a lot of excessive damage; additionally, it is also shown to boost the body supply of antiviral cytokines; (4) Shiitake: ability to stop the growth of the virus by preventing the entry into the cell; this mushroom has shown to be effective in fighting the herpes simplex virus, hepatitis C virus, HIV, and the influenza.

Mushrooms fight viral infections, and there are many studies on antiviral activities of several mushrooms against herpes (HHV-causing skin infections) [123], West Nile (mosquito-borne disease) [124], Orthpoxviruses (variola agent) [125], influenza [126], hepatitis B [127], and human immunodeficiency (HIV) [128]. The most studied mushroom strains for producing antiviral bioactive compounds include *Coriolus versicolor, Lentinula edodes, Grifola frondosa, Ganoderma lucidum, Hericium erinaceus, Pleurotus ostreatus, Cordyceps sinensis, Laricifomes officinalis, Lenzites betulina, Rozites caperata,* and *Daedaleopsis confragosa* [129].

People cannot avoid harmful bacteria and viruses, but they can become ill less often and with shorter periods if the immune system is strong. The objective of enhancing immunity is attractive, but the ability to do so has proved equivocal for several reasons. The immune system is precisely that: a complex network, not a single entity. There are no scientifically proven direct links between various lifestyle changes and enhanced immune function; nevertheless, one can boost the immune system by for example sleeping well in order to release protective cytokines, taking zinc [130], vitamins A, C, and E [131], curcumin (turmeric) [132], as well as consuming mushroom dietary supplements or fresh mushroom, therefore preventing the chance of contracting viral diseases [133].

*6.1. HIV/AIDS*

We have previously reviewed this subject [134], and the pathogenesis of the disease is considered multifactorial. Nutrition is a fundamental part of a comprehensive package of care for people living with HIV/AIDS, and mushrooms supply bioactive molecules that may help patients [110]. To cushion the repercussion of the disease, widespread, action taken must integrate all elements involved, including nutritional care [135]. To assess and reduce the severity of the complex interaction that HIV/AIDS and malnutrition have on each other, it is essential to forecast the evolution of the disease and the probability of morbidity and death toll [136].

Mushroom β-glucans increase CD4 cells production and stimulate the immune system macrophages. Even when infected with HIV, the macrophages fight effectively and reduce HIV replication [137]. Several triterpenes from *Ganoderma lucidum* are active as antiviral agents against human immunodeficiency virus type 1 (HIV-1) [138].

In addition to polysaccharides and triterpenoids displaying a variety of medicinal properties, mushrooms contain many antimicrobial factors, which include lentinan, ganaderiol-F, ganoderic acid-β, lucidumol, PSP, coprinol, campestrin, sparassol, armillaric acid, cortinellin, and ustilagic acid [139–141]. These active compounds fight viruses in two major ways: (a) they boost the immune system: directly (specific response) and/or through various factors of humoral and cellular immunity [142]; and (b) they attack the virus directly, which prevents the proliferation of viruses and can stop viral infections from developing [143].

Direct antiviral effects include inhibition of viral enzymes, synthesis of viral nucleic acids, and adsorption or uptake of viruses [144]. Indirect antiviral effects are achieved by stimulating the immune response against the viral invasion and promoting biochemical factors, such as alkalinity, which discourage viral replication [145]. In many mushrooms, β-glucans, glycoproteins, melanins, terpenoids, and nucleosides displayed antiviral activity [146].

Through the lymphotropic nature of the virus HIV-1, it infects humans, and the function of the lymph node is disrupted, the production of dendritic cells is increased, and there is accumulation in lymph nodes, presenting exogenous microbial antigens [147].

Drug resistance to anti-HIV drugs is emerging, and many people infected with HIV have serious adverse reactions. Antiviral compounds from mushrooms (e.g., triterpenes, phenolic compounds, ergosterol peroxide, and purine derivatives) are strong biotherapeutics acting directly on the pathways of enzymatic system of the human host, regulating the interactions between viral and components of the human cell [148].

They may also act by inhibiting viral enzymes carried within the capsid and on the viral envelope, while some are only produced in the infected cell [149]. The antiviral compounds of mushrooms also may condition the virus genome intervening on the synthesis pathway of viral nucleic acids and its penetration of viruses into cells [150].

### 6.2. Herpes Virus

The Herpes Simplex Virus (HSV-1) co-evolved with humans for thousands of years in a constant, dynamic, and endless dance where the pathogen is present at a high prevalence, affecting globally half of the human population [151,152].

While there are more than 100 known herpes viruses, two strains occur in most β-amyloid plaques of Alzheimer's Disease (AD), as their proteins are two-thirds identical, suggesting that this common virus may be a possible risk factor for AD, showing some evidence that specific viral species directly contribute to a risk of developing AD [153].

The neurotropic virus can either remain in a dormant state, with occasional revitalization events, or eventually originate severe acute encephalitis, which is marked by aggravated neuroinflammation and extended neuroimmune activation, producing a life-threatening neurological disease [154]. HSVs also alter host cell metabolism, inducing antiviral mechanisms and reprogramming cell death in non-immune cells; they are also capable of inducing apoptosis in immune cells and the death of T cells, while allowing viral replication to occur in epithelial cells before uprising into the neural ganglia, producing a latent infection [155].

Antiviral activity of the mycelia of higher mushrooms (*Pleurotus ostreatus*, *Fomes fomentarius*, *Auriporia aurea*, *Polyporus squamosus*, and *Coriolus versicolor*) against influenza virus type A (serotype H1N1) and herpes simplex virus type 2 (HSV-2) was determined to be effective [156]. They occur in SSA but may not be edible due to their texture and bitter taste but used as medicinal and functional properties [157].

### 6.3. Influenza Virus

Several mushrooms in natural form or as a food supplement are effective on preventing and treating a variety of viruses such as the common cold and the flu virus. This is significant upon considering the highly infectious nature and ability of these viruses to mutate. *Boletus edulis*, *Datronia molis*, *Calvatia gigantea*, *Laricifomes officinalis*, *Suillus luteus*, *Coriolus versicolor*, *Lentinus edodes*, *Lenzites betulina*, and *Piptoporus betulinus* were shown to be effective against the flu-causing influenza viruses [116,158]

### 6.4. Human Papillomaviruses (HPVs)

The use of *Coriolus versicolor* biomass supplement in women for 1 year revealed a great efficacy, whether in the regression of the cervical dysplasia (LSIL) or in the disappearance of the High-Risk HPV. This dietary supplementation showed positive therapeutic impact either in the reversion of LSIL (with High-Risk HPV+) or in those HSIL patients who have undergone surgery, but the High-Risk HPV viral count continued to increase [159].

This was subsequently replicated with active hexose correlated compound (AHCC), which is a fermented extract of cultured *Lentinula edodes* mycelia that is administered for at least 6 months with a 60% successful elimination of human papillomavirus (HPV) infections in women with positive PAP smears [160]. A recent study involving 42 patients showed that a combination of administration of *Coriolus versicolor* biomass provided positive outcomes in cases of primary or recurrent genital warts [161].

Mushroom biomass forms may be given as a complement in aggregation with surgery, chemo-, or radiotherapy, with a significant influence on NK cell activity when induced by the presence of a viral infection.

### 6.5. The Novel Coronavirus (SARS-CoV-2)

Currently, no specific treatment has been identified for COVID-19. The interesting thing about this SARS-CoV-2 virus is the symptoms, which can range from no conceivable symptoms all the way to having severe cases of all major symptoms, lower respiratory tract infection with fever, dry cough, and dyspnoea, spreading the virus. There are a vast number of studies that have been done with mushrooms as a potential antiviral treatment but very few yet specifically with this new virus [162]

Recently, *Cordyceps sinensis* and *Cordyceps militaris* were claimed be effective agents for the prevention and treatment of COVID-19 by immunomodulating, reducing the proinflammatory cytokines, preventing lung fibrosis, improving tolerance to hypoxemia, and inhibiting the viral enzymes [163]. *Lentinus edodes*, *Grifola frondosa*, and *Inonotus obliquus* are considered to have therapeutic potential as a natural antiviral treatment against SARS-COV-2, opening the research into this field.

In Norway, *Agaricus blazei*, *Ganoderma lucidum*, *Hericium erinaceus*, and *Grifola frondosa* were considered to have preventive or curative effect against the severe lung inflammation and acute pneumonia that often complicates COVID-19 infection [164].

A recent study in Iraq showed that *Ganoderma lucidum* uptake on some hematological and immunological response in patients with Covid-19 had a significant role in helping in the treatment of COVID-19 infections [165].

Mushrooms are the highest dietary source for the unique sulfur-containing antioxidant ergotheionine. This amino acid is a Generally Recognized as Safe (GRAS) product by the FDA and gets into the food chain mainly through mushroom consumption. There is a recent study revealing ergotheionine's potentially beneficial role in SARS-CoV-2 cases [113].

The above claims must not be generalized to the recent SARS-CoV-2 infection [166], and the immediate priority is to harness innate immunity to accelerate early antiviral immune responses.

## 7. Antitumour Activity of Mushrooms

Usually, the causes of cancer are multifactorial, and they include genetic, environmental, and other risk factors. A recent meta-analysis of 213 studies, including 77 clinical studies, showed that *Ganoderma lucidum* or *Coriolus versicolor* mushrooms enhanced the efficacy and ameliorated their adverse effects, which lead to an improved quality of life in cancer patients [108,167].

Mushroom lectins are a group of proteins/glycoproteins that can possess immunomodulating as well as direct cytotoxic activity toward tumour cell lines. In mushroom extracts and biomass, there are also some anticancer haemolysing proteins [168], enzyme laccase [169], ribosome-inactivating proteins [170], and ubiquitin-conjugated proteins, which also display direct cytotoxic activity [171,172].

Polysaccharides of mushrooms have antitumor activity, which is associated with the immunostimulatory effect that they can exert, since they activate foreign body reactions from the immune system [173]. This antitumor activity is not caused by a direct cytotoxic effect but via activation of the innate immune system of the host. The mechanism of action is related to the presence of pattern recognition receptors that can recognize the polysaccharides as pathogen-associated molecular patterns (PAMPs), due to its high molecular weight [174].

Consequently, proinflammatory cytokines are produced in a cascade, including tumour necrosis factor alpha (TNF-$\alpha$), which are members of the IL-1 family that regulate immune homeostasis and the mechanisms against infections in recognition of foreign cells and tumour cells [175].

Some structures of mushroom β-glucans are better adapted to specific receptors, which suggests a relationship between the structure and antitumor activity of polysaccharides, and it was found that mostly β-1,3-glucans have the highest antitumor activity [176,177]. Triterpenes, the secondary compounds found in mushrooms, cause tumour cells to self-destruct (apoptosis) [178,179].

Polysaccharide extracts from *Hericium erinaceus* are active against liver cancer cells in vitro and in vivo [180,181]. The highest consumption of dietary mushrooms, including *Agaricus bisporus* and *Lentinula edodes*, is associated with a decreased risk of breast cancer in premenopausal women and postmenopausal women [182].

Maitake mushroom (*Grifola frondosa*) is one of the most popular edible medicinal mushrooms. The natural killer (NK) cells, which have the ability to eliminate target cells without prior immunization, show an important role in controlling viral infections and high cytotoxic activity in oncologic patients administered *G. frondosa*, and they significantly restrain tumour growth. This is achieved by an increased release of TNF-α and IFN-γ from the spleen and a significant boost in IFN-γ and TNF-α expressed in NK cells [183].

## 8. Prebiotic Activity of Mushrooms

Prebiotics act as food for probiotics, and some health benefits of prebiotics, such as reducing glucose levels in the blood and improvement of the bowel function, have been medically proven and recognized by health authorities [184].

Endogenous β-glucans show better prebiotic properties than exogenous β-glucans. We have discussed the role of some bacteria responsible for the degradation of mixed linked β-glucans in the small intestine and in the hind gut [185].

Currently, inulin, fructo-oligosaccharides (FOS), galacto-oligosaccharides (GOS), lactulose, and polydextrose are recognized as the well-established prebiotics, but there is evidence that β-glucans can also be a source of long chain prebiotics [186]. *Pleurotus ostreatus* and *Pleurotus eryngii* have a potential stimulator effect on the growth of probiotic bacteria [187]. *Pleurotus* and *Cyclocybe* mushrooms, studied in terms of their prebiotic potential, exhibited a beneficial influence on the composition of gut microbiota of apparently healthy and elderly subjects [188].

The prebiotic effect of mushroom biomass (e.g., *Coriolus versicolor*) on human gut populations of total aerobes and anaerobes showed that dietary mushroom inclusion beneficially affected gut homeostasis performance and exerted changes in intestinal microbial communities [189].

## 9. Mushrooms and Neurological Disorders

One of the most challenging public health problems in Africa is data collection, and only very few epidemiological studies have been carried out in Sub-Saharan Africa, but generally, there is a high reported prevalence of neurological morbidity and disorders (e.g., epilepsy, dementia, stroke), which have been escalating [190,191].

When cells generate energy, they use oxygen and yield free radicals as a consequence of ATP generation by the mitochondria, which is a cell organelle that has a critical role in the development of neurodegenerative disorders [192]. The human body has various mechanisms to prevent oxidative stress by either yielding inner natural antioxidants (e.g., catalase, enzymes glutathione peroxidase, superoxide dismutase) or having them provided through foodstuffs and/or dietary supplements [193].

Neurodegeneration caused by disruptions of crucial homeostatic interactions between circulation and the brain may be mediated by microbial products that modulate the gut–brain axis, causing neuro-inflammation and neuronal dysfunction [194]. Neuro-inflammation can be caused by virus DNA/RNA infection, which challenges the host immune system, and continued exposure to the inflammatory mediators (e.g., cytokines, chemokines, and ROS) can result in neuronal dysfunction and degeneration [195].

People who incorporate mushrooms into their diets, even in small amounts (more than twice a week), seem to have a lower risk of mild cognitive impairment, usually

preceding Alzheimer's disease [196]. Mushrooms contain many other substances whose exact role in brain health is not yet clear, but they include hericenones, erinacines terpenoids, scabronines, isoindolinones, sterols, and dictyophorines, which are a series of compounds that could contribute to the growth of nerve and brain cells [197].

*Hericium erinaceus* has been studied as a precursor of acetylcholine, which has neuroprotective and anti-neurodegenerative properties. *H. erinaceus* mycelium shows great promise for the treatment of Alzheimer's and Parkinson's diseases [198].

We have previously discussed how abnormal redox homeostasis and oxidative stress causes diverse neuropsychiatric disorders and the immunomodulation role of mushroom biomass of *Coriolus versicolor* [199]. Presently, the interest is to focus on mediator markers of oxidative stress and neuroinflammation in progressive neurodegenerative disorders and distinct configurations of chronic mental illness [200].

Oxidative stress and altered antioxidant systems have been considered an important factor underlying the pathogenesis of Alzheimer's disease. Brain inflammation has been linked to many diseases, including amyotrophic lateral sclerosis (ALS), multiple sclerosis (MS), Parkinson's disease (PD) and, particularly, Alzheimer's disease (AD) [201].

We have previously discussed [186] the emerging role of lipoxin A4 and inflammasome in neurodegeneration and the potential therapeutic role of mushroom *Coriolus versicolor*. Integrated survival responses exist in the brain, which are under the control of redox-dependent genes, called vitagenes, including heat shock proteins (HSPs), sirtuins, thioredoxin, and lipoxin A4. The activation of LXA4 signalling and modulation of stress-responsive vitagene proteins could serve as a potential therapeutic target for AD-related inflammation and neurodegenerative damage [202,203].

Mushrooms through their powerful antioxidant characteristics have the potential to protect neurons in mitochondrial dysfunctions-associated aging and neurological disorders [204].

Anxiety symptoms and disorders, more common than depression, are among the most common primary care challenges in medical practice. L-theanine is an amino acid (an analogue of amino acids L-glutamate and L-glutamine) found most in tea leaves and in mushrooms. It should be noted that health claims for L-theanine as a supplement are not recognized in the European Union but are approved by the FDA. However, mushrooms, as well as dietary biomass supplements, not extracts, as food containing L-theanine do not need any approval, becoming a functional food secondary component of medical treatment [205].

*Coriolus versicolor*, a common healthful mushroom, has been receiving increasing attention by its antitumoral, anti-inflammatory, antioxidant, antibacterial, and immunomodulatory properties, including in the hippocampus. Our data unveiled a so far unexplored neurogenic potential of *Coriolus versicolor* supplementation as a possible preventive strategy for different neurological conditions [110,203].

*Role of Mushrooms in Autism*

According to the World Health Organization, one child in 270 worldwide suffers from an autism spectrum disorder. Autism is more common in Africa than initially believed, and it is a growing global public health concern. Most Africans are largely unaware of autism, which is a highly heritable neurodevelopmental disorder that is often confused with witchcraft, curses or spells, and demons, and children with autism in SSA tend to be diagnosed only around age 8, some 4 years later than worldwide.

Many African children with autism, more often boys than girls, are usually hidden away at home, and the prevalence is unknown, while only few clinicians have the skills or experience to identify the condition. Indeed, maternal and child mental health services have not been a priority, since child mortality and malnutrition are more urgent concerns [206,207].

Autism Spectrum Disorder (ASD) is a disorder still very poorly understood that is caused by genetic or environmental factors; it is first recognized in early childhood in the form of a multi organ system disability caused by impaired neurogenesis and apoptosis,

impaired synaptogenesis and synaptic pruning or an imbalanced excitatory–inhibition system [208].

A few studies with a handful of cases have been dealt with in South Africa, Nigeria, Ethiopia, and Kenya, fostering the conviction that ASD is more severe in African children than elsewhere with up to 4% of children having the condition [209]. Recent epidemiological studies revealed a possible important link between mycotoxin exposure and neurodevelopmental disorders with regard to ASD [210].

We have previously discussed dietary mushrooms and supplements, which have specific effects on gastrointestinal inflammation in ASD patients. The most commonly used mushrooms as potent health-boosters, which may bring some hope to autistic children and families, include Chaga (*Inonotus obliquus*), Reishi (*Ganoderma lucidum*), Turkey Tail (*Coriolus versicolor*), Shiitake (*Lentinula edodes*), Lion's Mane (*Hericium erinaceus*), Cordyceps (*Cordyceps militaris*), and oyster mushroom (*Pleurotus giganteous*). They have shown beneficial to symptoms relating to anxiety and depression, which are related to both autism and attention deficit–hyperactivity disorder (ADHD) [211].

The therapeutic potential of *Hericium erinaceus* bioactive and bioavailable components that pass the blood–brain barrier has been demonstrated [212]. They condition several functions, including triggering the production of nerve growth factor, the obstruction of the cytotoxicity of an extracellular heterogeneous mixture of small peptides plaque deposits, and the shielding against neuron lysis [213].

## 10. Concluding Remarks

The right to adequate food of acceptable quality has not been achieved in SSA. Hunger and malnutrition in the general SSA population remained a challenge even prior to the pandemic. COVID-19 is affecting food systems globally and has negative combined impacts on agricultural markets, economic recession, food insecurity, acute malnutrition, high levels of childhood illnesses, and water-borne diseases further threatening the health and life of people, namely children. Among the recovery measures to adopt, there is the need to maximize macro- and micronutrient intakes.

It is necessary to valorise indigenous foods and detect novel local food sources to aid and promote healthy lifestyle, income, wellness, and wellbeing. Authorities and international aid should aim helping the country's smallholder farmers make the transition move from subsistence to community commercial farming on a long-term ambitious plan.

Africa constitutes at least 25% of the total mushroom biodiversity in the world, but it has been barely researched. The abundant agricultural waste found in SSA offers opportunity for mushroom production. Mushrooms have distinct nutritional and bioactive profiles and have been absent in global dietary guidelines despite its high biological value. There is limited information about the nutritive, therapeutic, and biomedicinal uses of mushrooms in Africa. Mushrooms complement the human diet and microbiota requirements with various ingredients not found or deficient in food items of plants and animal origin, being considered an ultimate health food for the prevention of various human diseases.

Recognizing mushrooms as good sources of bioactive components, in strengthening human immune system, enhancing natural body resistance, and lowering proneness to disease with little scope of toxicity or overdose, along with their minimal side effects, make them ideal candidates for developing novel foods, dietary supplements, and therapies.

**Author Contributions:** Conceptualization, T.F.; validation, J.F. and C.G.; writing-original draft preparation, T.F.; writing-review and editing, J.F., V.B., C.G., and T.V.; supervision, T.F. and T.V. All authors have read and agreed to the published version of the manuscript.

**Funding:** This research received no external funding.

**Institutional Review Board Statement:** Not applicable.

**Informed Consent Statement:** Not applicable.

**Data Availability Statement:** Not applicable.

**Conflicts of Interest:** All authors declare no conflict of interests.

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
