# Peer review of "Mushroom Nutrition as Preventative Healthcare in Sub-Saharan Africa"

_applsci, doi:10.3390/app11094221_

Round 1

Reviewer 1 Report

The authors reviewed an important topic of diet quality in Sub-Saharan Africa. Based on the scientific evidence the authors intended to offer a new, African conditions-based food pyramid in which native mushroom species are one of the main sources of proteins. Besides, the authors aimed to offer an overview of health protection with mushroom nutraceuticals. Although, the goals of this paper are very up-to-date, with a fresh and previously underlooked source of nutrients like mushrooms the overall impression is that the paper seems disconnected. What is the paper’s topic? In the present form, it looks like there are two papers with two topics that only occasionally overlap. The main change in the offered pyramid is "mushrooms", and that is not obvious from the paper’s title. Moreover, the second part which deals with the health benefits of mushroom nutraceuticals has two major flaws: it is not connected with the food pyramid since it often discusses species that can not be found in African countries, thus, are hardly an integral part of people's diet. The title also says „nutraceuticals“, which Chang and Buswell defined in 1996 as products belonging to dietary supplements. More up-to-date definitions look at it even stricter, so nutraceuticals are extracts and prepared in the form of a pill or capsule. So, mushrooms in their native state can be marked as superfood, not nutraceuticals.

In the present form it is not clear which mushroom’s compounds are in focus: nutritive ones or medicinal; so how is the proposed food pyramid connected with the rest of the paper? Examples of species consumed in Sub-Saharan Africa should be discussed regarding the proposed food pyramid.

In the abstract, the authors stated that local increase in production and productivity (of mushrooms?) is imperative. This has not been addressed in the manuscript itself. If there are native species proven to be highly nutritious and health-protective they need to be discussed. Are they wild species or cultivated in Africa? Are they accessible to the African population? What is the benefit of highly valuable Hericium erinaceus if the population in Sub-Saharan Africa can not purchase it or find it in nature? The same goes for species effective against cancer and viruses. If we focus on the food pyramid where mushrooms are emphasized as new sources of proteins why the authors did not focus on proteins originating from mushrooms that exhibit health-promoting properties (instead of scattering all they have found on mushrooms)?

Eather the authors rewrite the entire manuscript and discuss how the species present in Sub-Saharan Africa can contribute to overall health when incorporated in the daily diet (and actually prove the benefits of the proposed pyramid) or write a general review about benefits from mushrooms as whole or mushroom nutraceuticals. But in the latter case, it is hard to expect that expensive preparations like mushroom nutraceuticals can become a daily routine in the region where living standard is much lower.

Additional, specific comments are given in the manuscript.

Author Response

Letter to Reviewers                                           Manuscript ID: applsci-1186196

First and foremost we thank you for all comments which we agree with all of them.

Perhaps I should explain the reason for this article and its extension. Following the publication of our last book “Food and Nutrition Security in Africa” (2020; ISBN. 978-989-8934-05-5) suggested by ICSU, and because the theme deserved expansion in other domains, we were allocated a Chapter on another Book requested by the International Science Council and the African Academy of Sciences. Unfortunately, and after our part was ready, the idea of publishing faded for several reasons.

We have worked towards the objective of providing Africans with at least one different perspective of how to change or improve the present panorama of malnutrition, knowing that this is presently unachievable and even worsen with the pandemic. International aid has not been the solution and searching for local products and unexplored foods was the aim of this new book.

We have been working with mushrooms for decades and we are part of a Working Group from the African Academy of Sciences with the objective of studying how to improve indigenous mushrooms, and how to establish or cultivate the well-known species from other parts of the world. The production of mushrooms on a small or large scale is quite simple and viable. Good examples can be seen in Cameroon, Congo, Ivory Coast, Kenya, and Namibia, where 10 kg of fresh mushrooms per square meter and more is presently obtained.

Even some considered “exotic” like Hericium exists in many parts of Africa. The sole exception seems to be (not yet reported) Cordyceps growing an entomopathogenic fungi.

This was the reason why the Keywords were chosen searching by using scientific sites such as Medline, PubMed, and Google Scholar, by all authors, and information gathered. We have inserted now appropriate keywords. Searches were conducted in English, French and Portuguese, languages we dominate.

In what refers the present submission:

The Title yes we decided it should be changed as per your query and to take away the topic of the pyramid as central role, and hope it is now reflecting the contents.

The dietary guidelines are not for kids but for general population. Nutrition was only established as an independent science in 2008 in Mozambique (we were involved in establishing the first course and lecture for 7 years) and very few nutritionists exist to support authorities in their decision. For kids we have followed the School Feeding Programme that Brazil established in 1954 and we have established only in 2013.

So the objectives were to enhance the positive role of African foods, as markets are flooded with imported and processed foods, while trying to flag the message of the continent's little exploited gold mine: mushrooms, as nutritive foods and as foods with biomedical value. In this regard, the topic expands to dietary mushroom supplements, still non-produced locally but to explain the hidden value of their bioactive elements.

We have deleted figure 2.

Agree that nutritional and dietary guidelines are totally different. We have inserted a 2021 Reference where it shows that only this year in the USA it is suggested the inclusion of mushrooms.

We have chosen a pyramid to reflect our ideas as this format was used in our lectures in Mozambique trying to explain the priorities and the disadvantages of some food ingredients.

The word Nutraceutical also caused confusion and we have tried to eliminate since presently there is still no agreement and legislation of this term. In the 80’s this term was also used as synonym as Pharmanutrient.

To note that dietary supplement of mushroom biomass are under the scope of novel foods in Europe, and do not need approval as they are considered part of foods. However, mushroom extracts, yes, they are medicinal products, in need to comply with medicinal legislation.

The suggestion of using the Brazilian NOVA classification was not taken into account as our topic did not aim at discussing food processing methods.

We have deleted the mention to 9 servings of fruit and vegetables a day not to cause confusion but the reference asked is from the USDA (current 2015–2020 Dietary Guidelines for Americans) that recommends eating five to nine servings of fruits and vegetables per day (2.5 cups of vegetables and about 2 cups of fruit per day.​ If you divide these numbers up throughout your meals, you’re looking at eating).

To refer to the size and number of servings in the African food guidelines, food consumption studies are required. There is a methodology proposed by the FAO but still a pilot Global Individual Food consumption data Tool. Through a web platform it will be possible to collect, harmonize and disseminate data available at national and sub-national level all over the world. But in SSA we are still far from joining this international and important task. EFSA in Europe are following this evaluation. However, if one wanted to be precise, the requirement would be to study also the microbiota of global populations, and we are still far from this stage.

We believe one cannot claim that many mushroom species cannot be found in African countries. Indeed, a recent study about biodiversity distribution indispensable for nature conservation and sustainable management of natural resources, covered mushrooms: more than 16,000 records of fungi representing 4843 species and infraspecific taxa were found in 860 publications relating to West Africa alone (Piepenbring, M., Maciá-Vicente, J.G., Codjia, J.E.I. et al. Mapping mycological ignorance – checklists and diversity patterns of fungi known for West Africa. IMA Fungus 11, 13 (2020). https://doi.org/10.1186/s43008-020-00034-y).The objective of this article is also to alert that numerous species of fungi are thought to remain undiscovered in tropical regions and biodiversity hotspots in SSA.

Immediate causes of deforestation is wild mushroom depletion that could lead to food insecurity among villagers. Young people and farmers must be coached on the values of indigenous crops including mushrooms.

Again, mushroom nutraceuticals are indeed made form extracts. Pills from biomass mushroom are, on the contrary, considered dietary supplements, not pharmaceuticals, yet still not adequate for African markets. The native species mushrooms proven to be highly desirable need to be further investigated as nutritious sources. But this even for developed countries, as much is still unknown. Efforts to develop modern based African mushroom meals have yet to be effective.

The lack of technical and scientific graduates limits the debate and there are very few laboratories that perform such tasks, the very first should in assessing fungal species richness, systematic positions, and ecology, in order to promote mycology and fungal conservation.

Thank you.

24th April 2021

Reviewer 2 Report

Attachment.

Author Response

(The authors gave the same response as above.)

Round 2

Reviewer 1 Report

Dear Authors,

Thank you! The comments you provided are extensive, grounded, and supported by explanations and references. In addition, you addressed all of them in a satisfactory manner. The new title better reflects this important topic. Indeed, not many research papers concerning mushrooms are coming from Africa.

Since authors are much involved in the topic I would like to add a comment which is not strictly scientific but might be good info for future development. I am familiar with the work of The Future of Hope foundation from Zimbabwe which produces mushrooms in local communities and empowers women through this business. As far as I know, they are introducing Nestle coffee waste into their production as a substrate for oyster mushroom cultivation. So, there are positive examples of changing the perspective on a local level, and maybe a connection with them might expand your idea with this pyramid and malnutrition problem.

Your present work is also informative and provided (me as well) with data about local species and potential. I see that as an additional value of this paper and strongly support its publishing in the present form.

I have only one comment/request for change. Think that you should change the subtitle 4.1. from Bioactive elements of mushrooms to Bioactive compounds of mushrooms.

As for the food pyramid, I still think that it is overcrowded with photos and not clearly represent the author's aim. However, I am not familiar with how people in Africa comment or understand it so I might be wrong.

Thank you again for your good work!

Author Response

Letter to Reviewers                                           Manuscript ID: applsci-1186196

First and foremost we thank you for all comments which we agree with all of them.

Perhaps I should explain the reason for this article and its extension. Following the publication of our last book “Food and Nutrition Security in Africa” (2020; ISBN. 978-989-8934-05-5) suggested by ICSU, and because the theme deserved expansion in other domains, we were allocated a Chapter on another Book requested by the International Science Council and the African Academy of Sciences. Unfortunately, and after our part was ready, the idea of publishing faded for several reasons.

We have worked towards the objective of providing Africans with at least one different perspective of how to change or improve the present panorama of malnutrition, knowing that this is presently unachievable and even worsen with the pandemic. International aid has not been the solution and searching for local products and unexplored foods was the aim of this new book.

We have been working with mushrooms for decades and we are part of a Working Group from the African Academy of Sciences with the objective of studying how to improve indigenous mushrooms, and how to establish or cultivate the well-known species from other parts of the world. The production of mushrooms on a small or large scale is quite simple and viable. Good examples can be seen in Cameroon, Congo, Ivory Coast, Kenya, and Namibia, where 10 kg of fresh mushrooms per square meter and more is presently obtained.

Even some considered “exotic” like Hericium exists in many parts of Africa. The sole exception seems to be (not yet reported) Cordyceps growing an entomopathogenic fungi.

This was the reason why the Keywords were chosen searching by using scientific sites such as Medline, PubMed, and Google Scholar, by all authors, and information gathered. We have inserted now appropriate keywords. Searches were conducted in English, French and Portuguese, languages we dominate.

In what refers the present submission:

The Title yes we decided it should be changed as per your query and to take away the topic of the pyramid as central role, and hope it is now reflecting the contents.

The dietary guidelines are not for kids but for general population. Nutrition was only established as an independent science in 2008 in Mozambique (we were involved in establishing the first course and lecture for 7 years) and very few nutritionists exist to support authorities in their decision. For kids we have followed the School Feeding Programme that Brazil established in 1954 and we have established only in 2013.

So the objectives were to enhance the positive role of African foods, as markets are flooded with imported and processed foods, while trying to flag the message of the continent's little exploited gold mine: mushrooms, as nutritive foods and as foods with biomedical value. In this regard, the topic expands to dietary mushroom supplements, still non-produced locally but to explain the hidden value of their bioactive elements.

We have deleted figure 2.

Agree that nutritional and dietary guidelines are totally different. We have inserted a 2021 Reference where it shows that only this year in the USA it is suggested the inclusion of mushrooms.

We have chosen a pyramid to reflect our ideas as this format was used in our lectures in Mozambique trying to explain the priorities and the disadvantages of some food ingredients.

The word Nutraceutical also caused confusion and we have tried to eliminate since presently there is still no agreement and legislation of this term. In the 80’s this term was also used as synonym as Pharmanutrient.

To note that dietary supplement of mushroom biomass are under the scope of novel foods in Europe, and do not need approval as they are considered part of foods. However, mushroom extracts, yes, they are medicinal products, in need to comply with medicinal legislation.

The suggestion of using the Brazilian NOVA classification was not taken into account as our topic did not aim at discussing food processing methods.

We have deleted the mention to 9 servings of fruit and vegetables a day not to cause confusion but the reference asked is from the USDA (current 2015–2020 Dietary Guidelines for Americans) that recommends eating five to nine servings of fruits and vegetables per day (2.5 cups of vegetables and about 2 cups of fruit per day.​ If you divide these numbers up throughout your meals, you’re looking at eating).

To refer to the size and number of servings in the African food guidelines, food consumption studies are required. There is a methodology proposed by the FAO but still a pilot Global Individual Food consumption data Tool. Through a web platform it will be possible to collect, harmonize and disseminate data available at national and sub-national level all over the world. But in SSA we are still far from joining this international and important task. EFSA in Europe are following this evaluation. However, if one wanted to be precise, the requirement would be to study also the microbiota of global populations, and we are still far from this stage.

We believe one cannot claim that many mushroom species cannot be found in African countries. Indeed, a recent study about biodiversity distribution indispensable for nature conservation and sustainable management of natural resources, covered mushrooms: more than 16,000 records of fungi representing 4843 species and infraspecific taxa were found in 860 publications relating to West Africa alone (Piepenbring, M., Maciá-Vicente, J.G., Codjia, J.E.I. et al. Mapping mycological ignorance – checklists and diversity patterns of fungi known for West Africa. IMA Fungus 11, 13 (2020). https://doi.org/10.1186/s43008-020-00034-y).The objective of this article is also to alert that numerous species of fungi are thought to remain undiscovered in tropical regions and biodiversity hotspots in SSA.

Immediate causes of deforestation is wild mushroom depletion that could lead to food insecurity among villagers. Young people and farmers must be coached on the values of indigenous crops including mushrooms.

Again, mushroom nutraceuticals are indeed made form extracts. Pills from biomass mushroom are, on the contrary, considered dietary supplements, not pharmaceuticals, yet still not adequate for African markets. The native species mushrooms proven to be highly desirable need to be further investigated as nutritious sources. But this even for developed countries, as much is still unknown. Efforts to develop modern based African mushroom meals have yet to be effective.

The lack of technical and scientific graduates limits the debate and there are very few laboratories that perform such tasks, the very first should in assessing fungal species richness, systematic positions, and ecology, in order to promote mycology and fungal conservation.

Letter to Reviewers                                           Manuscript ID: applsci-1186196

First and foremost we thank you for all comments which we agree with all of them.

Perhaps I should explain the reason for this article and its extension. Following the publication of our last book “Food and Nutrition Security in Africa” (2020; ISBN. 978-989-8934-05-5) suggested by ICSU, and because the theme deserved expansion in other domains, we were allocated a Chapter on another Book requested by the International Science Council and the African Academy of Sciences. Unfortunately, and after our part was ready, the idea of publishing faded for several reasons.

We have worked towards the objective of providing Africans with at least one different perspective of how to change or improve the present panorama of malnutrition, knowing that this is presently unachievable and even worsen with the pandemic. International aid has not been the solution and searching for local products and unexplored foods was the aim of this new book.

We have been working with mushrooms for decades and we are part of a Working Group from the African Academy of Sciences with the objective of studying how to improve indigenous mushrooms, and how to establish or cultivate the well-known species from other parts of the world. The production of mushrooms on a small or large scale is quite simple and viable. Good examples can be seen in Cameroon, Congo, Ivory Coast, Kenya, and Namibia, where 10 kg of fresh mushrooms per square meter and more is presently obtained.

Even some considered “exotic” like Hericium exists in many parts of Africa. The sole exception seems to be (not yet reported) Cordyceps growing an entomopathogenic fungi.

This was the reason why the Keywords were chosen searching by using scientific sites such as Medline, PubMed, and Google Scholar, by all authors, and information gathered. We have inserted now appropriate keywords. Searches were conducted in English, French and Portuguese, languages we dominate.

In what refers the present submission:

The Title yes we decided it should be changed as per your query and to take away the topic of the pyramid as central role, and hope it is now reflecting the contents.

The dietary guidelines are not for kids but for general population. Nutrition was only established as an independent science in 2008 in Mozambique (we were involved in establishing the first course and lecture for 7 years) and very few nutritionists exist to support authorities in their decision. For kids we have followed the School Feeding Programme that Brazil established in 1954 and we have established only in 2013.

So the objectives were to enhance the positive role of African foods, as markets are flooded with imported and processed foods, while trying to flag the message of the continent's little exploited gold mine: mushrooms, as nutritive foods and as foods with biomedical value. In this regard, the topic expands to dietary mushroom supplements, still non-produced locally but to explain the hidden value of their bioactive elements.

We have deleted figure 2.

Agree that nutritional and dietary guidelines are totally different. We have inserted a 2021 Reference where it shows that only this year in the USA it is suggested the inclusion of mushrooms.

We have chosen a pyramid to reflect our ideas as this format was used in our lectures in Mozambique trying to explain the priorities and the disadvantages of some food ingredients.

The word Nutraceutical also caused confusion and we have tried to eliminate since presently there is still no agreement and legislation of this term. In the 80’s this term was also used as synonym as Pharmanutrient.

To note that dietary supplement of mushroom biomass are under the scope of novel foods in Europe, and do not need approval as they are considered part of foods. However, mushroom extracts, yes, they are medicinal products, in need to comply with medicinal legislation.

The suggestion of using the Brazilian NOVA classification was not taken into account as our topic did not aim at discussing food processing methods.

We have deleted the mention to 9 servings of fruit and vegetables a day not to cause confusion but the reference asked is from the USDA (current 2015–2020 Dietary Guidelines for Americans) that recommends eating five to nine servings of fruits and vegetables per day (2.5 cups of vegetables and about 2 cups of fruit per day.​ If you divide these numbers up throughout your meals, you’re looking at eating).

To refer to the size and number of servings in the African food guidelines, food consumption studies are required. There is a methodology proposed by the FAO but still a pilot Global Individual Food consumption data Tool. Through a web platform it will be possible to collect, harmonize and disseminate data available at national and sub-national level all over the world. But in SSA we are still far from joining this international and important task. EFSA in Europe are following this evaluation. However, if one wanted to be precise, the requirement would be to study also the microbiota of global populations, and we are still far from this stage.

We believe one cannot claim that many mushroom species cannot be found in African countries. Indeed, a recent study about biodiversity distribution indispensable for nature conservation and sustainable management of natural resources, covered mushrooms: more than 16,000 records of fungi representing 4843 species and infraspecific taxa were found in 860 publications relating to West Africa alone (Piepenbring, M., Maciá-Vicente, J.G., Codjia, J.E.I. et al. Mapping mycological ignorance – checklists and diversity patterns of fungi known for West Africa. IMA Fungus 11, 13 (2020). https://doi.org/10.1186/s43008-020-00034-y).The objective of this article is also to alert that numerous species of fungi are thought to remain undiscovered in tropical regions and biodiversity hotspots in SSA.

Immediate causes of deforestation is wild mushroom depletion that could lead to food insecurity among villagers. Young people and farmers must be coached on the values of indigenous crops including mushrooms.

Again, mushroom nutraceuticals are indeed made form extracts. Pills from biomass mushroom are, on the contrary, considered dietary supplements, not pharmaceuticals, yet still not adequate for African markets. The native species mushrooms proven to be highly desirable need to be further investigated as nutritious sources. But this even for developed countries, as much is still unknown. Efforts to develop modern based African mushroom meals have yet to be effective.

The lack of technical and scientific graduates limits the debate and there are very few laboratories that perform such tasks, the very first should in assessing fungal species richness, systematic positions, and ecology, in order to promote mycology and fungal conservation.

Thank you.

24th April 2021

Reviewer 2 Report

I believe that you have made an excellent effort to improve your proposal and it now offers a better quality and technical consistency of valuable contributions.

Author Response

(The authors gave the same response as above.)
